# Mitigating oxygen loss to improve the cycling performance of high capacity cation-disordered cathode materials

Jinhyuk Lee[1], Joseph K. Papp [2], Raphaële J. Clément [1], Shawn Sallis [3], Deok-Hwang Kwon[1], Tan Shi[1], Wanli Yang [3], Bryan D. McCloskey[2,4] & Gerbrand Ceder[1,5]

Recent progress in the understanding of percolation theory points to cation-disordered lithium-excess transition metal oxides as high-capacity lithium-ion cathode materials. Nevertheless, the oxygen redox processes required for these materials to deliver high capacity can trigger oxygen loss, which leads to the formation of resistive surface layers on the cathode particles. We demonstrate here that, somewhat surprisingly, fluorine can be incorporated into the bulk of disordered lithium nickel titanium molybdenum oxides using a standard solid-state method to increase the nickel content, and that this compositional modification is very effective in reducing oxygen loss, improving energy density, average voltage, and rate performance. We argue that the valence reduction on the anion site, offered by fluorine incorporation, opens up significant opportunities for the design of high-capacity cation-disordered cathode materials.

[1] Department of Materials Science and Engineering, University of California, Berkeley, CA 94720, USA. [2] Department of Chemical and Biomolecular Engineering, University of California, Berkeley, CA 94720, USA. [3] Advanced Light Source, Lawrence Berkeley National Laboratory, Berkeley, CA 94720, USA. [4] Energy Storage and Distributed Resources Division, Lawrence Berkeley National Laboratory, Berkeley, CA 94720, USA. [5] Materials Science Division, Lawrence Berkeley National Laboratory, Berkeley, CA 94720, USA. Correspondence and requests for materials should be addressed to G.C. (email: gceder@berkeley.edu)

With ever increasing demand for high-performance lithium-ion batteries, cathode materials with high energy density have been sought from diverse chemical spaces[1–3]. Recently, Li-excess disordered (rocksalt) transition metal oxides (LEX-RS), such as $Li_{1.211}Mo_{0.467}Cr_{0.3}O_2$, $Li_{1.3}Mn_{0.4}Nb_{0.3}O_2$, $Li_{1.2}Mn_{0.4}Ti_{0.4}O_2$, and $Li_{1.2}Ni_{0.333}Ti_{0.333}Mo_{0.133}O_2$, have gained much attention as a new class of high-capacity cathode materials, delivering capacities as high as 300 mAh g$^{-1}$ [4–12]. The surge of interest in these materials, previously thought to be inactive[13–15], is due to the realization that cation disorder can be tolerated as long as enough Li excess is present in the compound. When the Li content, $x$, in $Li_xTM_{2-x}O_2$ (TM=transition metal) is greater than about 1.1 (10% Li excess), Li diffusion can occur through a percolating network of so called 0-TM channels, which are considerably less sensitive to the reduction in local O–O distance caused by disorder than the 1-TM channels in layered materials (Fig. 1a)[4, 5]. Removing the constraint that ions form well-ordered layered structures opens up a large, previously unexplored chemical space of metal oxides.

One issue with LEX-RS materials is that the Li excess reduces the TM content and thereby increases the average TM oxidation state, decreasing the TM-based redox capacity unless the TM can be oxidized up to $TM^{5+}$ or $TM^{6+}$ [2, 6, 16, 17]. In addition, both Li excess and cation disorder create linear Li–O–Li configurations in which the O 2p orbitals do not hybridize with a TM orbital[17]. The higher energy of these labile O 2p states makes them easier to oxidize, bringing them in competition with TM oxidation upon delithiation[16, 17]. The combined effect of an enhanced O redox process and limited TM content often results in small TM-based redox capacities in disordered Li-excess materials, and the remaining capacity is delivered by O redox processes when these materials achieve high capacity (Fig. 1a)[6–10].

While reversible oxygen oxidation is an exciting direction to create high-capacity materials, high levels of oxygen redox can trigger oxygen loss from the Li-excess materials[2, 8–10]. The cations left behind by this process create cation-densified surface phases with poor Li-transport kinetics as their lowered Li contents puts them below the threshold of 0-TM percolation (Fig. 1a)[2, 9]. In addition, oxygen species evolved from the surface can react with the electrolyte to form a resistive layer at the surface of the cathode particles, increasing the impedance of the cell[18, 19]. As a result, LEX-RS cathodes that experience severe oxygen loss have shown limited cyclability[8–10].

This issue has for instance been studied in detail for cation-disordered Li–Ni–Ti–Mo oxides. Materials such as $Li_{1.15}Ni_{0.375}Ti_{0.375}Mo_{0.1}O_2$ and $Li_{1.2}Ni_{0.333}Ti_{0.333}Mo_{0.133}O_2$ deliver high capacities, yet <60% of their ~200 mAh g$^{-1}$ theoretical Ni-based capacity is used upon cycling[9]. Instead, extensive use of the O redox reservoir leads to severe oxygen loss near the surface of the cathode particles, which in turn results in large polarization of the voltage profile and degrades their cycling performance[9, 17]. We note that similar behavior is observed in several other LEX-RS cathodes, such as Li–Ti–Fe oxides and Li–Nb–M oxides (M=Ni, Co, Fe)[6, 8, 10]. Therefore, for all of these materials, reducing oxygen loss is a crucial step towards further performance improvements.

In this work, we present a simple but surprisingly effective strategy to reduce oxygen loss from LEX-RS materials and thereby significantly increase their practical energy density. Based on a comparative study of $Li_{1.15}Ni_{0.375}Ti_{0.375}Mo_{0.1}O_2$ (LN15), $Li_{1.2}Ni_{0.333}Ti_{0.333}Mo_{0.133}O_2$ (LN20), and $Li_{1.15}Ni_{0.45}Ti_{0.3}Mo_{0.1}O_{1.85}F_{0.15}$ (LNF15), we demonstrate that, in contrast to layered oxides, oxygen can be partially substituted by fluorine in Li-excess cation-disordered rocksalts using a standard solid-state method. By lowering the average anion valence, fluorine incorporation enables an increase of (redox active) $Ni^{2+}$ content per formula unit of the Li–Ni–Ti–Mo oxides from 0.375 Ni and 0.333 Ni for LN15 and LN20, respectively, to 0.45 Ni for LNF15,

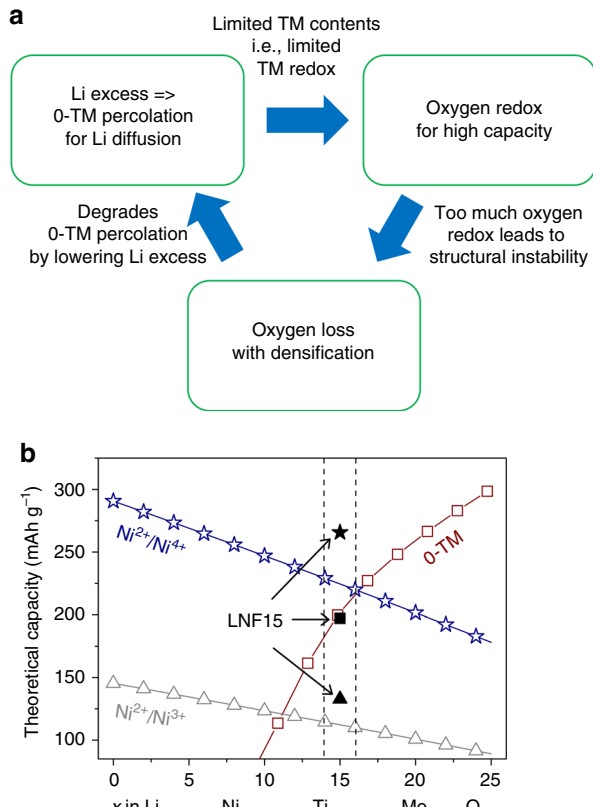

**Fig. 1** Material design principles. **a** Relationship between Li excess, oxygen redox and oxygen loss. Oxygen loss needs to be prevented so as to ensure a percolating network of 0-TM (transition metal) channels allowing facile Li diffusion in Li-excess disordered materials[9]. **b** Theoretical capacity of $Li_{1.15}Ni_{0.45}Ti_{0.3}Mo_{0.1}O_{1.85}F_{0.15}$ (LNF15) overlaid with that of a range of Li–Ni–Ti–Mo oxides. Three different capacities, computed under the assumption of various possible mechanisms, are plotted as a function of Li-excess level: $Ni^{2+}/Ni^{4+}$ redox capacity (star), $Ni^{2+}/Ni^{3+}$ redox capacity (triangle), and 0-TM capacity (square) that is defined as the Li capacity accessible through a percolating 0-TM network in the disordered rocksalt structure[4, 5]. Black symbols correspond to the fluorinated compound

leading to a higher theoretical $Ni^{2+}/Ni^{4+}$ capacity for LNF15 (266 mAh g$^{-1}$) than for LN15 (225 mAh g$^{-1}$) and LN20 (202 mAh g$^{-1}$) (Fig. 1b). Therefore, charge compensation in LNF15 relies to a greater extent on Ni redox processes and less heavily on O redox, leading to reduced oxygen loss. As a result, fluorine-substituted LNF15 cycles with smaller polarization at reasonable operating voltages and exhibits a much higher energy density (790 Wh kg$^{-1}$, 3330 Wh l$^{-1}$) and rate capability than the two pure Li–Ni–Ti–Mo oxides (LN15, LN20).

## Results

**Synthesis and characterization.** All materials (LN15, LN20, and LNF15) were successfully synthesized using a standard solid-state method. This is in contrast to previous studies on the fluorination of layered or disordered rocksalt-type cathodes, which either failed to incorporate fluorine in the bulk lattice of layered materials via solid-state synthesis[20, 21], or employed mechanochemical ball milling to synthesize fluorinated disordered rocksalt phases[11]. X-ray diffraction (XRD) (Fig. 2a) and elemental analysis (Table 1) indicate that the compositions of the as-synthesized disordered rocksalt phases are close to the target compositions. XRD refinements show that the lattice parameter slightly increases with Li excess, from 4.1444 Å (LN15: 15% Li excess) to 4.1449 Å (LN20:

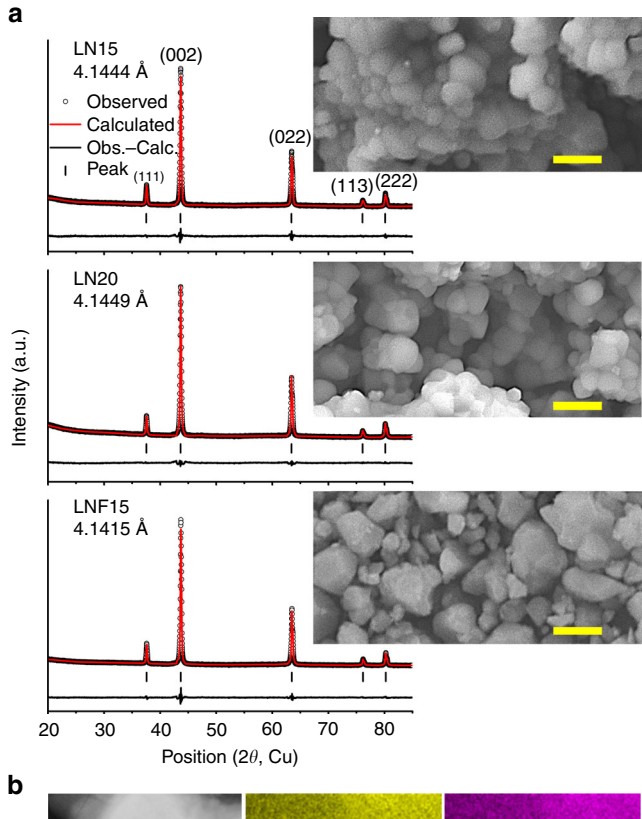

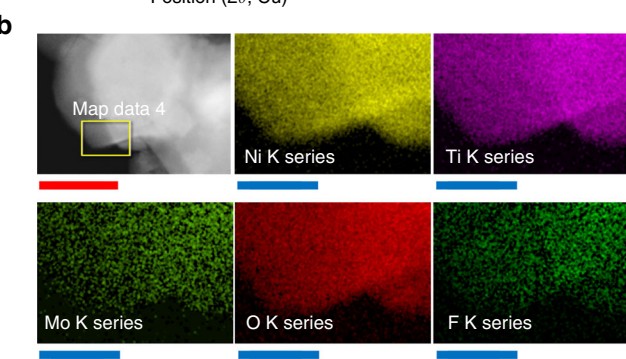

**Fig. 2** Structure characterization of $Li_{1.15}Ni_{0.375}Ti_{0.375}Mo_{0.1}O_2$ (LN15), $Li_{1.2}Ni_{0.333}Ti_{0.333}Mo_{0.133}O_2$ (LN20), and $Li_{1.15}Ni_{0.45}Ti_{0.3}Mo_{0.1}O_{1.85}F_{0.15}$ (LNF15). **a** X-ray diffraction patterns and refinement results of LN15, LN20, and LNF15. The inset shows the scanning electron microscopy image of each material. Scale bars, 200 nm. **b** Energy dispersive spectroscopy mapping (Ni, Ti, Mo, O, F) on one area (yellow square) of a LNF15 particle. Scale bars, red: 100 nm, blue: 25 nm

20% Li excess), but decreases to 4.1415 Å (LNF15) with fluorine substitution (Supplementary Table 1).

Scanning electron microscopy (SEM) shows that the average primary particle size of LNF15 (~180 nm) is larger than that of LN15 (~100 nm) and LN20 (~ 100 nm). The average particle size of LNF15 was subsequently decreased to ~50 nm, using a shaker mill (S-LNF15) and to ~70 nm via high-energy ball-milling (H-LNF15) (Supplementary Fig. 1).

To confirm that fluorine is substituted in the bulk disordered lattice instead of forming secondary phases at the surface of the particles, we first performed energy dispersive spectroscopy using a transmission electron microscope (TEM-EDS) (Fig. 2b). EDS mapping reveals a uniform distribution of fluorine in the LNF15 particle, along with other elements. $^{19}F$ solid-state nuclear magnetic resonance spectroscopy (ssNMR) experiments were subsequently performed to prove that the fluorine distribution observed with TEM-EDS is not due to LiF on the surface of the particles, as often

**Table 1 Target vs. measured Li: Ni: Ti: Mo: F atomic ratio of $Li_{1.15}Ni_{0.375}Ti_{0.375}Mo_{0.1}O_2$ (LN15), $Li_{1.2}Ni_{0.333}Ti_{0.333}Mo_{0.133}O_2$ (LN20) and $Li_{1.15}Ni_{0.45}Ti_{0.3}Mo_{0.1}O_{1.85}F_{0.15}$ (LNF15) compounds by direct current plasma emission spectroscopy and an ion selective electrode**

| Materials | Target Li: Ni: Ti: Mo: F | Measured Li: Ni: Ti: Mo: F |
|---|---|---|
| LN15 | 1.15: 0.375: 0.375: 0.1: 0 | 1.138: 0.39: 0.369: 0.103: 0 |
| LN20 | 1.2: 0.333: 0.333: 0.1333: 0 | 1.18: 0.35: 0.332: 0.137: 0 |
| LNF15 | 1.15: 0.45: 0.3: 0.1: 0.15 | 1.13: 0.464: 0.305: 0.1: 0.142 |

found in layered oxides[20, 21]. NMR spectra were collected on both LNF15 and LiF powders (Fig. 3). The $^{19}F$ spin echo NMR spectra obtained for LNF15 and LiF exhibit significant differences, and the $^{19}F$ pj-MATPASS spectrum[22] (in which spinning sidebands are suppressed, allowing us to extract the chemical shifts of the different $^{19}F$ environments in the sample) collected on LNF15 consists of a number of broad, overlapping signals spanning a wide range of resonant frequencies (Fig. 3). The $^{19}F$ NMR data reveal the presence of multiple fluorine local environments in LNF15, in contrast to the single fluorine site observed for LiF, confirming that F is doped into the bulk Li–Ni–Ti–Mo oxide lattice. The various LNF15 signals are shifted and significantly broadened as compared with the unique LiF resonance, presumably as a result of paramagnetic interactions between the $^{19}F$ nuclei and unpaired electrons coming from nearby $Ni^{2+}$ cations, suggesting that F and Ni are within a few angstroms (<5 Å). Consistent with this interpretation, the different F sites observed in LNF15 can be assigned to $^{19}F$ nuclei with varying numbers of $Ni^{2+}$ cations in their first, second, and/or third metal coordination shells (Supplementary Fig. 2a). Fits of the different spectra shown in Fig. 3 are presented, and the various fluorine sites present in LiF and LNF15 are discussed, in the Supplementary Information (Supplementary Fig. 2b, Supplementary Table 2, and Supplementary Note 1). The combined TEM-EDS and $^{19}F$ ssNMR results clearly indicate that fluorine substitution in a disordered rocksalt can be achieved using a standard solid-state method, and that the target LNF15 phase is successfully made.

**Electrochemical properties.** In Fig. 4, the electrochemical properties of LN15, LN20, LNF15, and shaker-milled LNF15 (S-LNF15) are compared using galvanostatic cycling between 1.5 and 4.6 V. At 20 mA g$^{-1}$, LN15 and LN20 deliver high discharge capacities up to 194 mAh g$^{-1}$ (587 Wh kg$^{-1}$, 2454 Wh l$^{-1}$) and 220 mAh g$^{-1}$ (672 Wh kg$^{-1}$, 2775 Wh l$^{-1}$), respectively (Fig. 4a, b). However, a large fraction of the capacity is delivered below 2.5 V, resulting in the average discharge voltage of ~3.03 V and ~3.05 V for LN15 and LN20, respectively. In particular, a discharge plateau at ~2.2 V is seen after the 3.8 V plateau, as was observed for other Ni-redox-based cation-disordered oxides, such as $Li_{1.167}Ni_{0.25}Ti_{0.583}O_2$[6, 23, 24]. Previously, we showed that this 2.2 V plateau accompanies reduction of $Ti^{4+}$ and $Mo^{6+}$ at the particle surface, which becomes possible after oxygen loss[9]. In addition, their voltage profiles exhibit a large hysteresis (voltage gap, polarization) between charge and discharge. On the contrary, LNF15 and S-LNF15 cycle with much reduced polarization, delivering high discharge capacities up to 210 mAh g$^{-1}$ (681 Wh kg$^{-1}$, 2894 Wh l$^{-1}$) and 250 mAh g$^{-1}$ (790 Wh kg$^{-1}$, 3334 Wh l$^{-1}$), respectively (Fig. 4c, d). The reduced polarization results in an increase in the average discharge voltage to ~3.2 V for both LNF15 and S-LNF15. In addition, the 2.2 V discharge plateau is hardly seen for LNF15 and S-LNF15.

To more directly compare the polarization for LN15, LN20, and LNF15, their first-cycle and second-charge voltage profiles are overlaid in Fig. 5a. The profiles of LN15 and LN20 feature

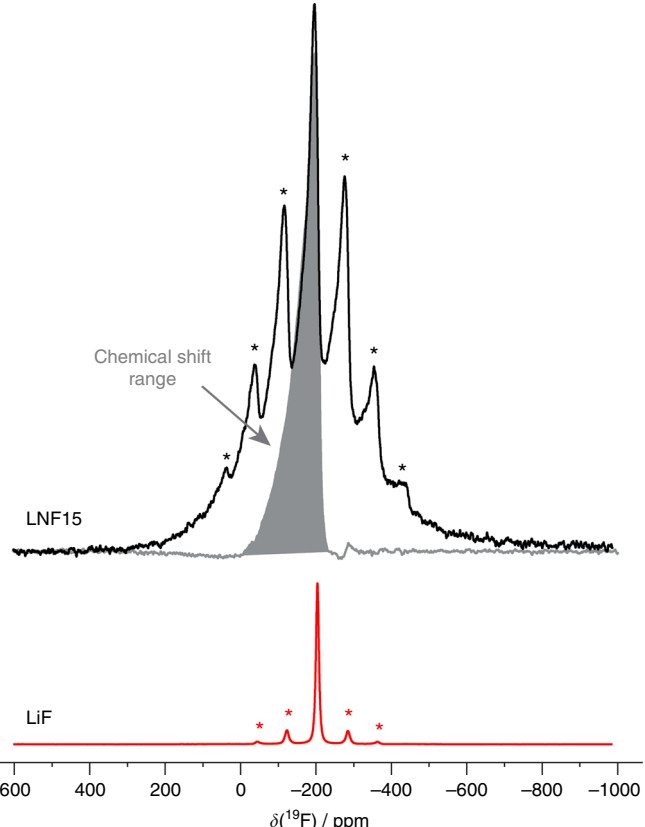

**Fig. 3** $^{19}$F spin echo nuclear magnetic resonance (NMR) spectra obtained at 30 kHz magic-angle spinning (MAS) for Li$_{1.15}$Ni$_{0.45}$Ti$_{0.3}$Mo$_{0.1}$O$_{1.85}$F$_{0.15}$ (LNF15) and LiF. Spinning sidebands in the spin echo spectra are indicated with an asterisk. The $^{19}$F projected magic-angle turning phase-adjusted sideband separation (pj-MATPASS) isotropic spectrum collected on LNF15, shown with a gray shading, reveals the presence of broad, overlapping signals spanning a wide range of resonant frequencies

large voltage gaps in the middle of charge and discharge, and the 2.2 V first discharge plateau is poorly recovered on subsequent charge. On the other hand, the smaller voltage gap and the absence of an asymmetric 2.2 V plateau for LNF15 indicate that fluorination decreases the polarization, resulting in a considerably larger discharge capacity for LNF15 and S-LNF15 than for LN15 or LN20 above 2.3 V (Figs. 4 and 5a). For example, above 3 V (2.5 V), S-LNF15 delivers 60 mAh g$^{-1}$ (80 mAh g$^{-1}$) more discharge capacity than LN15.

We conducted galvanostatic intermittent titration tests (GITT) to gain insight into the origin of the reduced polarization for the fluorinated sample. Figure 5b compares the voltage–capacity GITT profiles obtained on discharge for LNF15 and LN20 (after a first charge of 250 mAh g$^{-1}$): the inset shows the voltage–time GITT profiles around 210 h, corresponding to a discharge capacity of ~120 mAh g$^{-1}$ in the voltage–capacity plot. The GITT profiles show considerably less voltage relaxation in LNF15 than in LN20, especially in the middle of discharge. The voltage–time profiles in the inset show that the time-dependent portion of voltage relaxation is much smaller in LNF15 than in LN20. This data indicates that the mass-transfer resistance is decreased in LNF15[25, 26].

The positive effect of fluorination is further confirmed by the reasonably good rate capability of LNF15 after decreasing its particle size through shaker milling. Figure 5c shows the discharge profiles of S-LNF15, when charged at 20 mA g$^{-1}$ and discharged at different rates between 1.5 and 4.6 V. As the discharge rate

increases from 10 to 20, 40, 100, 200, and 400 mA g$^{-1}$, the capacity of S-LNF15 decreases from 261 mAh g$^{-1}$ (819 Wh kg$^{-1}$) to 234 mAh g$^{-1}$ (752 Wh kg$^{-1}$), 221 mAh g$^{-1}$ (728 Wh kg$^{-1}$), 208 mAh g$^{-1}$ (677 Wh kg$^{-1}$), and 194 mAh g$^{-1}$ (549 Wh kg$^{-1}$), respectively. At the highest rate, the capacity of S-LNF15 is 50% higher than that of LN15 (~130 mAh g$^{-1}$) (Supplementary Fig. 3).

**Gas evolution measurement.** In order to determine whether reduced oxygen loss accounts for the better performance of the fluorinated material, differential electrochemical mass spectrometry (DEMS) measurements were performed on LN15, LN20, and LNF15 during cycling between 1.5 and 4.8 V at 20 mA g$^{-1}$ (Fig. 6a–c).

DEMS results indicate that LNF15 indeed experiences less oxygen loss than the un-substituted compounds. Upon first charge to 4.8 V, O$_2$ gas is detected from ~4.35 V (~185 mAh g$^{-1}$) for both LN15 and LN20, while O$_2$ gas detection is delayed to above ~4.5 V (~220 mAh g$^{-1}$) for LNF15. Moreover, the total amount of O$_2$ gas produced during initial charge decreases from 0.26 and 0.40 µmol mg$^{-1}$ (µmol of gas species per mg of active material) for LN15 and LN20, respectively, to 0.07 µmol mg$^{-1}$ for LNF15. Given that all of the O$_2$ gas very likely originates from the cathode compound, the amount of O$_2$ evolved corresponds to the loss of 2.3, 3.5, and 0.7% of the total oxygen content for LN15, LN20, and LNF15, respectively. Clearly, the amount of oxygen loss (LN20>LN15>LNF15) is inversely proportional to the Ni content (LN20<LN15<LNF15). Note that O$_2$ and CO$_2$ evolution also appears to occur at the very beginning of discharge, although most of this gas likely evolved during charge and required a few additional gas sampling pulses to completely sweep out of the cell headspace. Considering this, the total amount of O$_2$ evolved further increases to 0.30, 0.49, and 0.09 µmol mg$^{-1}$ for LN15, LN20, and LNF15, respectively.

For all samples, CO$_2$ evolves above ~4.4 V. Similarly to O$_2$ evolution, the amount of CO$_2$ evolved is smaller for LNF15 (0.05 µmol mg$^{-1}$) than for LN15 (0.14 µmol mg$^{-1}$) or LN20 (0.10 µmol mg$^{-1}$). Previous reports have shown that oxygen radicals generated from Li-excess materials upon first charge can attack carbonate-based electrolytes to produce CO$_2$ gas[16, 19]. Therefore, although the exact amount of CO$_2$ evolved according to this mechanism is unclear, as conventional electrolyte decomposition at a high voltage can also produce CO$_2$[27], some of the oxygen involved in CO$_2$ evolution might also be a manifestation of oxygen loss from the three compounds. Nevertheless, fluorine substitution leading to a higher Ni content clearly results in reduced gas evolution in these materials.

DEMS results on S-LNF15 with a smaller average particle of ~50 nm are shown in Supplementary Fig. 4. Upon first charge, ~0.11 µmol mg$^{-1}$ of both O$_2$ and CO$_2$ is evolved from the material, which is more than for LNF15 but still 44 and 55% less than for LN15 and LN20, respectively. Overall, it is clear that the reduced oxygen loss (or gas evolution) from LNF15, as compared with LN15 or LN20, is not merely a surface area effect but the consequence of fluorine substitution. Note that there is no F$_2$ gas evolution from LNF15 upon cycling (Supplementary Fig. 5).

**Structural changes upon cycling.** To understand how fluorine substitution and an increased Ni content affect the structural changes taking place upon cycling, in situ XRD was performed on LN20 and LNF15 as they were charged and discharged at 10 mA g$^{-1}$ between 1.5 and 4.6 V. The XRD patterns, voltage profiles and refined lattice parameters of LN20 and LNF15 are shown in Fig. 7a–d.

For both LN20 and LNF15, the (002) peak shift to a higher angle upon charge (indicating a decrease of the lattice parameter) is recovered on discharge, though both samples end up having

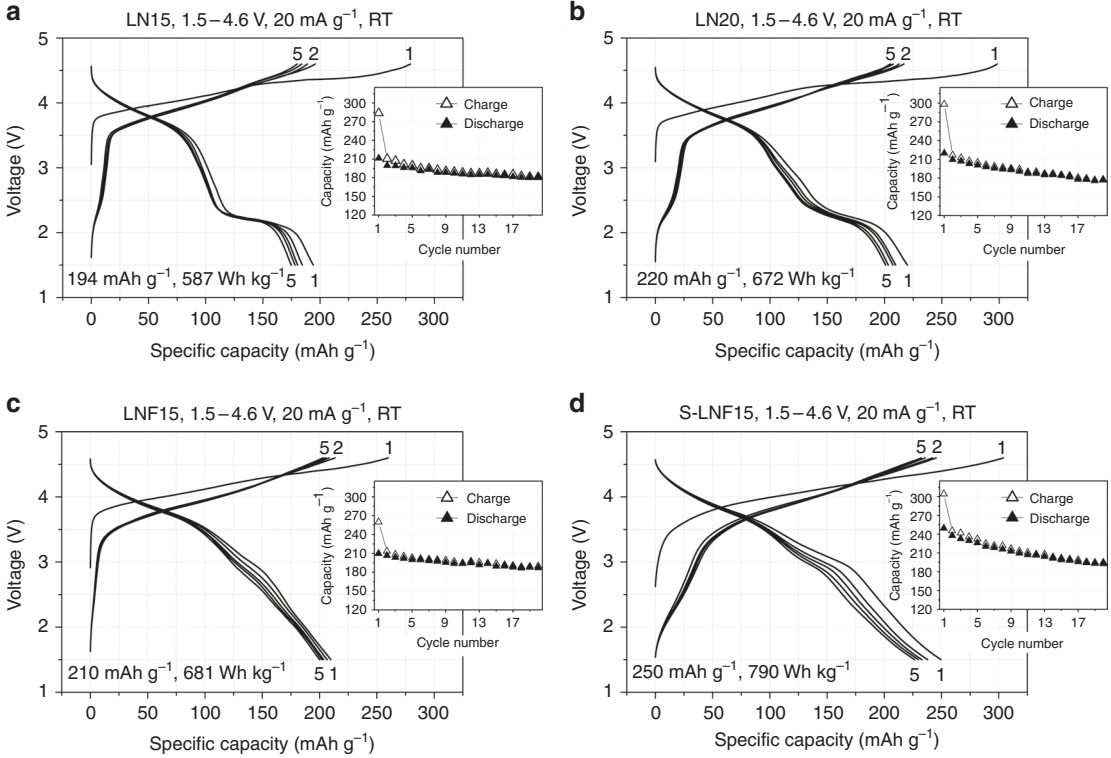

**Fig. 4** Cycling performance of $Li_{1.15}Ni_{0.375}Ti_{0.375}Mo_{0.1}O_2$ (LN15), $Li_{1.2}Ni_{0.333}Ti_{0.333}Mo_{0.133}O_2$ (LN20), and $Li_{1.15}Ni_{0.45}Ti_{0.3}Mo_{0.1}O_{1.85}F_{0.15}$ (LNF15). Voltage profiles of **a** LN15, **b** LN20, **c** LNF15, and **d** shaker-milled LNF15 (S-LNF15) when cycled between 1.5 and 4.6 V at 20 mA g$^{-1}$ at room temperature. The inset shows the capacity retention of the materials over the first 20 cycles

slightly larger lattice parameters at the end of discharge (LN20: 4.145 Å=>4.103 Å=>4.171 Å; and LNF15: 4.141 Å=>4.102 Å=>4.158 Å). The corresponding volume changes are −3.0% (+5.0%) upon first charge (discharge) for LN20, and −2.8% (+4.1%) upon first charge (discharge) for LNF15. The net volume increase after the first cycle (indicative of structural reorganization due to metal migration or oxygen loss) is smaller for LNF15 (+1.24%) than for LN20 (+1.87%), further confirming the higher reversibility of LNF15.

The major difference in behavior of the two materials is observed in the middle of the first charge. For LN20, the lattice parameter barely changes between ~120 and ~215 mAh g$^{-1}$ of charge. On the other hand, this behavior is not observed as clearly for LNF15. A lack of lattice parameter change is typically ascribed to a charging process that does not lead to oxidation of any species in the crystal structure, such as the combined extraction of lithium and oxygen[9]. Therefore, the extended region of negligible peak shift in the in situ XRD patterns of LN20 is consistent with the DEMS results, suggesting greater oxygen loss from LN20 than from LNF15 (Fig. 6). $O_2$ gas evolution from LN20 is in fact detected by DEMS upon charging past ~185 mAh g$^{-1}$ (~4.35 V), which overlaps with the region of negligible peak shift, albeit with some delay. The delay may be explained by oxygen loss that first creates intermediate species, such as $Li_2O$ or $Li_2O_2$, instead of producing $O_2$ gas molecules directly, as previously reported for layered Li-excess metal oxide cathodes[28].

**Redox mechanism.** Consistent with our initial hypothesis, fluorine substitution paired with the increased Ni-content results in reduced oxygen loss from the cathode. Assuming that the fractional utilization of the total Ni redox reservoir is ~50% (as in LN15 or LN20), the higher Ni-content of LNF15 requires less

oxygen redox to achieve a given capacity, decreasing the driving force for O loss.

To determine whether fluorine substitution affects the actual redox mechanism (e.g., the fractional utilization of the Ni or O redox reservoirs), soft X-ray absorption spectroscopy (sXAS) data was acquired on high-energy ball-milled LNF15 (H-LNF15) in both "bulk-sensitive" total fluorescence yield (TFY) mode, with a probe depth of 100–200 nm, and "surface-sensitive" total electron yield (TEY) mode, with a probe depth of about 10 nm. The electrochemical performance of H-LNF15 is shown in Supplementary Fig. 6. The Ni L-edge, O K-edge, and Ti L-edge spectra are shown in Fig. 8. There were no visible spectral features in the Mo M-edge. We expect Mo to remain largely as Mo$^{6+}$ (as previously observed for LN20)[9, 17], though a small amount of oxygen loss during the first charge may allow for some Mo reduction near the surface of LNF15 particles upon subsequent discharge[9, 29]. The F K-edge spectra are not presented due to the strong but non-relevant contribution from the PTFE binder. We expect F to remain as F$^-$ throughout cycling, since its high electronegativity (4.0) makes F oxidation unfavorable.

Figure 8a, b shows the Ni L-edge spectra of H-LNF15 in TFY and TEY mode, respectively. Because TM L-edges correspond to the excitation of 2p electrons to unoccupied 3d orbitals, they are direct probes of TM-3d states and allow to monitor changes in oxidation states[30, 31]. In general, TM L-edge sXAS features can be divided into two regions: the L$_3$-edge at lower excitation energy and the L$_2$-edge at higher energy, separated by the 2p core hole spin–orbit splitting[30, 31].

The Ni L-edge spectra obtained in bulk-sensitive TFY mode (Fig. 8a) reveal that, prior to cycling, the Ni L$_3$-edge region consists of a major peak at ~852.5 eV and a weak shoulder at ~855 eV. This is a typical lineshape of Ni$^{2+}$-containing electrode samples[31]. Such Ni$^{2+}$ lineshape persists for all the electrodes in

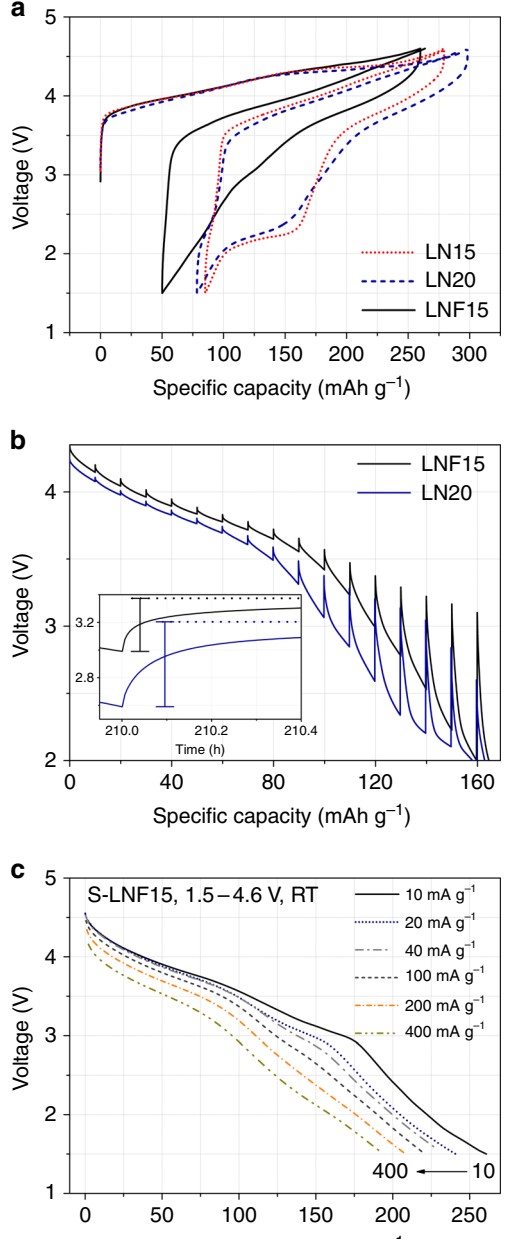

**Fig. 5** Electrochemical analysis of the origin of the improved cycling performance upon fluorine substitution. **a** Voltage profiles obtained upon first-cycle and second-charge: $Li_{1.15}Ni_{0.375}Ti_{0.375}Mo_{0.1}O_2$ (LN15: red dot), $Li_{1.2}Ni_{0.333}Ti_{0.333}Mo_{0.133}O_2$ (LN20: blue dash), and $Li_{1.15}Ni_{0.45}Ti_{0.3}Mo_{0.1}O_{1.85}F_{0.15}$ (LNF15: black solid). **b** First discharge voltage profiles of LN20 (blue) and LNF15 (black) from galvanostatic intermittent titration tests (GITT) after initial charge to 270 mAh g$^{-1}$: the inset shows the voltage–time GITT profiles around 120 mAh g$^{-1}$. Each step in GITT curve corresponds to a galvanostatic (dis)charge of an additional 10 mAh g$^{-1}$ at a rate of 20 mA g$^{-1}$ and is followed by a five-hour relaxation step. **c** Discharge profiles of shaker-milled LNF15 (S-LNF15), when charged at 20 mA g$^{-1}$ and discharged at different rates of 10, 20, 40, 100, 200, and 400 mA g$^{-1}$ (1.5-4.6 V, 298 K)

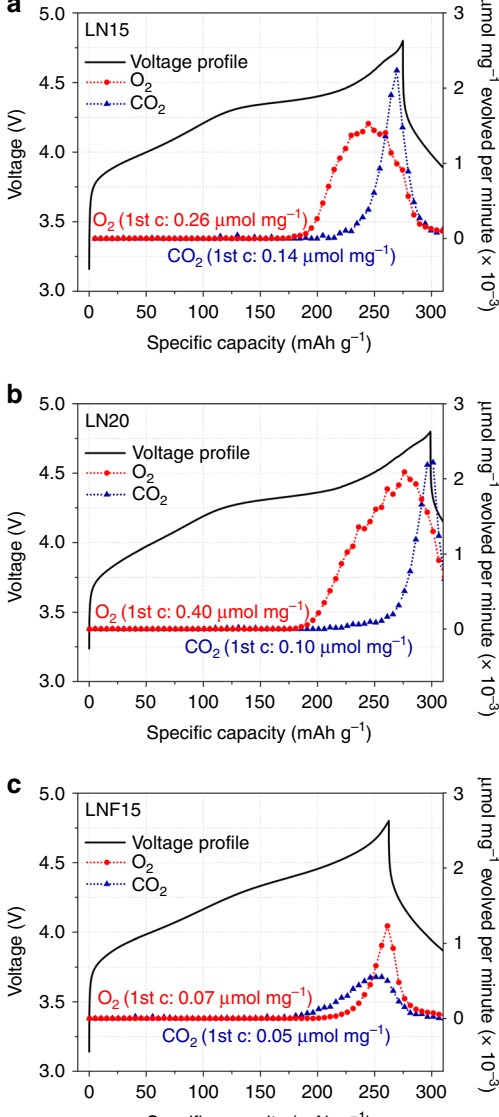

**Fig. 6** Differential electrochemical mass spectrometry (DEMS) study of $Li_{1.15}Ni_{0.375}Ti_{0.375}Mo_{0.1}O_2$ (LN15), $Li_{1.2}Ni_{0.333}Ti_{0.333}Mo_{0.133}O_2$ (LN20), and $Li_{1.15}Ni_{0.45}Ti_{0.3}Mo_{0.1}O_{1.85}F_{0.15}$ (LNF15). Voltage profiles (black solid) of **a** LN15, **b** LN20, and **c** LNF15 when charged to 4.8 V and discharged to 1.5 V at 20 mA g$^{-1}$, along with the DEMS results on $O_2$ (red circle) and $CO_2$ (blue triangle)

the surface-sensitive (~10 nm) TEY spectra (Fig. 8b), indicating the presence of electrochemically inactive Ni$^{2+}$ at the surface. This could either result from surface densification, as previously observed in layered compounds[2], or from surface reactions with the carbonate-based electrolyte, as mentioned earlier[19, 32].

The most obvious Ni-L TFY lineshape evolution upon electrochemical cycling is seen through the intensity ratio

between the peaks at 852.5 eV and the shoulder at 855 eV. The ratio decreases upon charging up to 4.6 V (~280 mAh g$^{-1}$) and returns to the ratio obtained for pristine H-LNF15 upon discharge to 1.5 V (~225 mAh g$^{-1}$). Correspondingly, the spectral weight of the Ni L$_2$-edge at ~873 eV increases upon charge and decreases upon discharge. This overall lineshape evolution of both the L$_3$- and L$_2$-edges is a typical sXAS evidence of Ni$^{2+}$ in the bulk of LNF15 particles oxidized to Ni$^{3+}$ and Ni$^{4+}$ upon charge, and reduced back to Ni$^{2+}$ on discharge[30, 31]. Overall, changes in the Ni L-edges are small compared with other Ni systems[31], and the Ni$^{2+}$ signal remains strong even after charging to 4.6 V, suggesting that only a small portion of the Ni redox reservoir is utilized in LNF15 during cycling. Additionally, because the main sXAS feature of Ni$^{4+}$ sits at higher energy compared with that of Ni$^{2+}$ and Ni$^{3+}$[31], the weak spectral weight above 855 eV indicates that most Ni has not been oxidized to Ni$^{4+}$ in LNF15. This behavior is similar to that of LN20 discussed earlier, for which

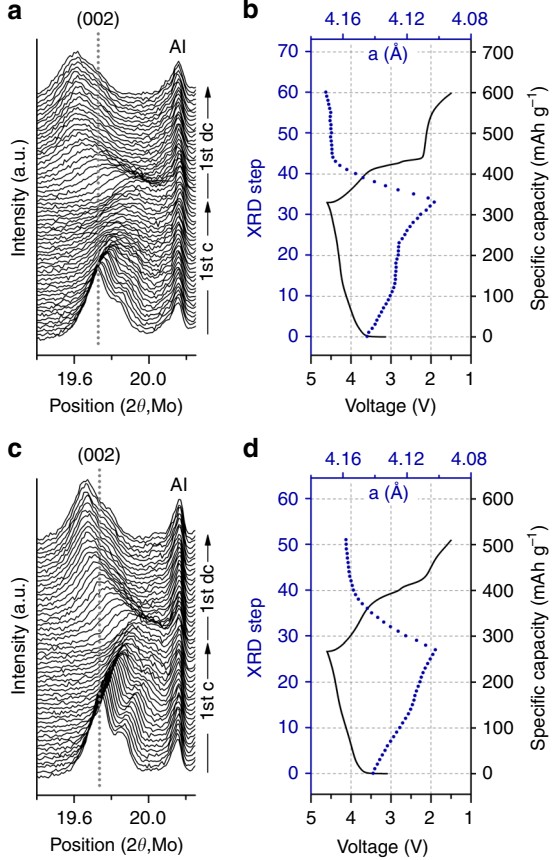

**Fig. 7** Structure evolution of $Li_{1.2}Ni_{0.333}Ti_{0.333}Mo_{0.133}O_2$ (LN20) and $Li_{1.15}Ni_{0.45}Ti_{0.3}Mo_{0.1}O_{1.85}F_{0.15}$ (LNF15) upon first charge (1st c) and discharge (1st dc). **a** In situ X-ray diffraction (XRD) patterns, **b** voltage profile (black line), and the lattice parameter (blue dot) of LN20. **c** In situ XRD patterns, **d** voltage profile (black line), and the lattice parameter (blue dot) of LNF15. In situ XRD cells were cycled at 10 mA g$^{-1}$ at room temperature between 1.5 and 4.6 V

hard XAS data revealed that the average Ni oxidation state does not go beyond Ni$^{3+}$ on charge[9].

Consistent with the Ni L-edge data presented above, O K-edge spectra (Fig. 8c, d) reveal that bulk Ni$^{2+}$ ions are oxidized on charge, but surface Ni remains inactive. The focus here is on "pre-edge" features below 534 eV in the O K-edge sXAS spectrum, most of which correspond to transitions from O-1s state to hybridized O-2p/TM-3d states. Figure 8c shows the O K-edge spectra in TFY mode. Particularly, the ~528.5 eV feature, which grows upon charge and disappears on discharge, indicates the presence of Ni$^{3+}$ or Ni$^{4+}$ hybridized with the O-2p states[31]. The observed peak growth at ~528.5 eV is therefore ascribed to partial Ni$^{2+}$/Ni$^{3+,4+}$ oxidation in the bulk, with extraction of Ni-dominated $e_g^*$ electrons, which is accompanied by some charge removal from the O-2p states due to the covalent nature of the $e_g^*$ states[31, 33–37]. In addition, we observe an increase in the overall pre-edge intensity in the 530–535 eV range (particularly at ~532.2 eV) upon delithiation, and most substantially after charging beyond 210 mAh g$^{-1}$ (Fig. 8c, Supplementary Fig. 7). An intensity gain at ~532 eV was previously assigned to O-hole creation in the non-bonding O 2p states in Li-excess materials, such as $Li_{1.2}Ni_{0.13}Co_{0.13}Mn_{0.54}O_2$[16, 38]. Therefore, the observed intensity gain in the 530–535 eV range likely indicates oxygen oxidation in LNF15. However, because most O K-edge features stem from hybridization of the TM-3d and O-2p states, the intensity depends on the strength of hybridization that is often

much enhanced upon delithiation[33, 34]. Therefore, it is unclear at this point how much of the intensity increase on charge is due to the introduction of holes in the non-bonding O 2p states[16, 17]. Nevertheless, the fact that a large amount of Ni ions remain in the Ni$^{2+}$ state upon charge suggests that oxygen oxidation is partly responsible for the large reversible capacity observed for H-LNF15.

Figure 8d shows the O K-edge spectra in TEY mode. Unlike the O K-edge spectra in TFY mode, there is negligible peak growth at ~528.5 eV upon charge, which is consistent with the Ni L-edge results and indicates that Ni$^{2+}$ at the surface remains inactive during electrochemical cycling. Furthermore, an overall lineshape change of the O-K TEY spectra is observed after 70 mAh g$^{-1}$ charge and is presumably due to a thin layer of electrolyte decomposition product coating the electrode surface at high potentials.

Negligible changes are observed in the Ti L-edge spectra of H-LNF15 in both TFY and TEY mode (Fig. 8e, f). In addition, nearly identical Ti L-edge spectra are obtained with TEM electron energy loss spectroscopy (EELS) on as-prepared LNF15 and LN20 (Supplementary Fig. 8). Based on a previous study, where we showed using DFT calculations and hard XAS that Ti is in its Ti$^{4+}$ state in LN20[9], we conclude that Ti in LNF15 remains as Ti$^{4+}$ throughout cycling.

In summary, the Ni$^{2+}$/Ni$^{3+,4+}$ redox reservoir is only partially utilized in in LNF15, which suggests that O redox processes have to be responsible for the high reversible capacity, as previously observed for LN20. A small amount of fluorination does not seem to affect the fractional utilization of the Ni redox reservoir, but the higher Ni content in LNF15 leads to a higher Ni-based capacity, as compared with LN15 or LN20.

## Discussion

Cation-disordered Li-excess metal oxides are a fascinating class of high energy density cathode materials. However, because of their need for Li excess, the high capacity often relies on oxygen redox processes which can lead to surface oxygen loss[8–10]. This in turn creates high-impedance layers at the surface of the particles, which degrades performance, in particular by reducing the average discharge voltage and rate capability. In this paper, we demonstrated that fluorine substitution is an effective means of mitigating this problem. As fluorine lowers the average anion valence, more Ni$^{2+}$ can be incorporated; not only increasing the Ni redox reservoir, but also preventing the compound from utilizing too much oxygen redox that can trigger oxygen loss (Fig. 1b). Un-substituted Li–Ni–Ti–Mo oxides deliver high capacities but exhibit large polarization after being charged above ~4.3 V (Supplementary Fig. 9)[9]. The fluorine-substituted compound, on the other hand, has reduced polarization, a higher average voltage and a considerably higher capacity above 2.5 V. DEMS and GITT measurements directly provide evidence that less oxygen is released, leading to smaller polarization near the middle of discharge. Indeed, upon charge to 4.8 V, O$_2$ gas evolution per gram of LNF15 is found to be ~3.6 times less than for LN15 and ~5.5 times less than for LN20, confirming our hypothesis that oxygen is less active in the fluorinated material.

The homogeneous incorporation of F into the material was confirmed with XRD, EDS, and NMR. Fluorine incorporation into the disordered Li–Ni–Ti–Mo–O rocksalt by conventional solid-state synthesis is maybe somewhat surprising as most attempts to prepare fluorinated layered-rocksalt oxides by solid-state synthesis resulted in the formation of LiF at the surface of the particles[20, 21]. The different behavior towards fluorine incorporation between the LEX-RS materials and well-ordered layered compounds may be related to the large variety of local environments that occur in cation-disordered materials as compared to the single local oxygen

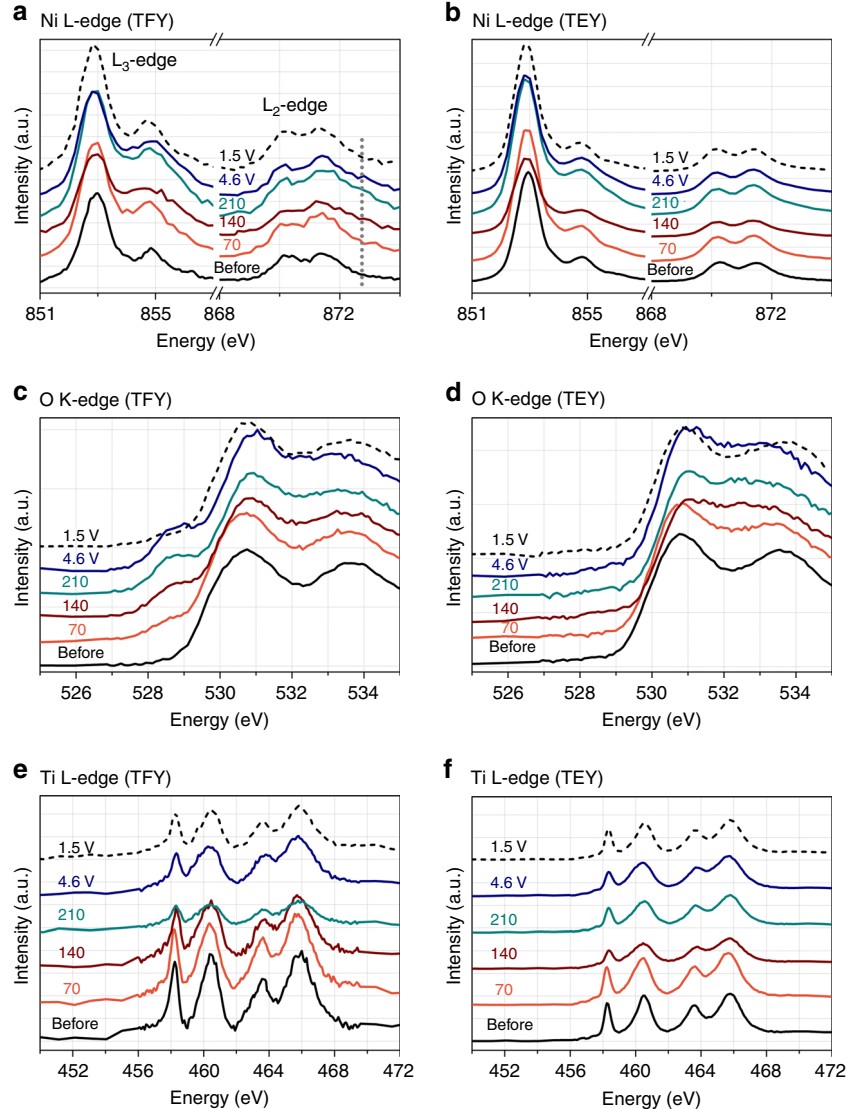

**Fig. 8** Soft X-ray absorption spectroscopy (sXAS) study of high-energy ball-milled $Li_{1.15}Ni_{0.45}Ti_{0.3}Mo_{0.1}O_{1.85}F_{0.15}$ (H-LNF15). sXAS spectra at the Ni L-edge: **a** bulk-sensitive total fluorescence yield (TFY) mode, **b** surface-sensitive total electron yield (TEY) mode. sXAS spectra at the O K-edge: **c** TFY mode, **d** TEY mode. sXAS spectra at the Ti L-edge: **e** TFY mode, **f** TEY mode. The data was collected before cycling H-LNF15 (black), after charging to 70 mAh g$^{-1}$ (orange), 140 mAh g$^{-1}$ (dark red), 210 mAh g$^{-1}$ (turquoise), 4.6 V (~ 280 mAh g$^{-1}$, dark blue), and after discharging to 1.5 V (black dash). H-LNF15 cells were cycled at 20 mA g$^{-1}$ at room temperature

environment in a layered oxide. We did find that there is a limit to the equilibrium incorporation of F, as evidenced by our failed attempt to synthesize $Li_{1.2}Ni_{0.4}Ti_{0.4}O_{1.6}F_{0.4}$, which resulted in a LiF secondary phase as observed with XRD (Supplementary Fig. 10). Using sXAS, we found that small levels of fluorine substitution do not lead to significant changes in the overall redox mechanism: as in LN20, the $Ni^{2+}/Ni^{4+}$ redox reservoir is still incompletely used in LNF15, and some additional capacity is provided by O-redox processes, as evidenced by Ni L- and O K-edge spectroscopy. Therefore, consistent with our initial hypothesis, the reduced O loss from LNF15 is most likely due to the increased $Ni^{2+}$ content in LNF15, which leads to an increased Ni-based capacity. Thus, compared to LN15 and LN20, charge compensation in LNF15 relies to a lesser extent on O redox, decreasing the driving force for O loss.

The reduction of oxygen loss is important as it leads to less cation densification at the particle surface. This is consistent with the smaller voltage polarization we find for LNF15 and S-LNF15, as compared with LN15 or LN20. Many disordered rocksalt materials achieve their large capacity from a discharge to very low

voltage[8–11]. As a result of fluorine substitution; we achieve an important increase of average discharge voltage, significantly increasing the practical value of the cathode material. To the best of our knowledge, S-LNF15 is the first disordered Ni-based compound that has more than 150 mAh g$^{-1}$ capacity at a voltage above 3 V. With 790 Wh kg$^{-1}$ (~3330 Wh l$^{-1}$), the energy content of S-LNF5 is higher than that of the un-fluorinated materials [LN15: ~590 Wh kg$^{-1}$ (~2470 Wh l$^{-1}$), LN20: ~670 Wh kg$^{-1}$ (~2790 Wh l$^{-1}$)], and compares well with that of existing commercial cathode materials. We find furthermore that fluorine substitution improves capacity retention as well as rate performance. Hence, although further optimization of the synthetic process is necessary to improve long-term stability and rate capability, S-LNF15 is a promising cation-disordered cathode material with Ni as the major TM-redox species.

In conclusion, based on a comparative study of $Li_{1.15}Ni_{0.375}Ti_{0.375}Mo_{0.1}O_2$, $Li_{1.2}Ni_{0.333}Ti_{0.333}Mo_{0.133}O_2$, and $Li_{1.15}Ni_{0.45}Ti_{0.3}Mo_{0.1}O_{1.85}F_{0.15}$, we demonstrated that fluorine can be incorporated into the bulk of disordered rocksalt oxides via a

solid-state method and that improving the content of redox active metals (e.g., Ni) with fluorination is an efficient strategy to reduce oxygen loss from the oxide lattice, leading to substantial performance improvements for cation-disordered Li-excess cathode materials. Overall, compositional modification based on fluorine substitution opens up opportunities for the design of high-capacity cathode materials containing TM species (Ni, Fe, Co, etc.) that were previously deemed unfavorable due to extensive overlap between the metal and oxygen redox states, leading to severe oxygen loss upon high delithiation.

## Methods

**Synthesis**. To synthesize $Li_{1.15}Ni_{0.375}Ti_{0.375}Mo_{0.1}O_2$ (LN15), $Li_{1.2}Ni_{0.333}Ti_{0.333}Mo_{0.133}O_2$ (LN20), and $Li_{1.15}Ni_{0.45}Ti_{0.3}Mo_{0.1}O_{1.85}F_{0.15}$ (LNF15), $Li_2CO_3$ (Alfa Aesar, ACS, 99% min), $NiCO_3$ (Alfa Aesar, 98%), $TiO_2$ (Anatase, Alfa Aesar, 99.9%), $MoO_2$ (Alfa Aesar, 99%), and LiF (Alfa Aesar, 99.99%) were used as precursors. A stoichiometric amount of precursors were dispersed into acetone and ball milled for 15 h, and then dried overnight in an oven. The mixture of the precursors was pelletized and then calcined at 750 °C for 2 h in air for $Li_{1.15}Ni_{0.375}Ti_{0.375}Mo_{0.1}O_2$ and $Li_{1.2}Ni_{0.333}Ti_{0.333}Mo_{0.133}O_2$, and 700 °C for 10 h for $Li_{1.15}Ni_{0.45}Ti_{0.3}Mo_{0.1}O_{1.85}F_{0.15}$, followed by furnace cooling to room temperature. After the calcination, the pellets were manually ground into fine powder.

**Electrochemistry**. To prepare a cathode film from untreated LN15, LN20, and LNF15, the powder of active compounds and carbon black (Timcal, SUPER C65) were first mixed manually using a mortar and pestle for 30 min. To make a cathode film from shaker-milled $Li_{1.15}Ni_{0.45}Ti_{0.3}Mo_{0.1}O_{1.85}F_{0.15}$ (S-LNF15), 700 mg of LNF15 and 200 mg of carbon black were first shaker-milled for 1 h in a (argon-filled) 45 ml zirconia vial with 10 g of 5 mm-diameter yttria stabilized zirconia balls (Inframat Advanced Materials) as grinding media, using SPEX 8000M Mixer/Mill. For high-energy ball-milled $Li_{1.15}Ni_{0.45}Ti_{0.3}Mo_{0.1}O_{1.85}F_{0.15}$ (H-LNF15), 280 mg of LNF15 and 80 mg of carbon black were first shaker-milled for 1 h in (argon-filled) stainless jars (50 ml) and mixed for 6 h at 400 rpm with three 10-mm-diameter and 15 3-mm-diameter stainless balls as grinding media, using a planetary ball mill (Retsch PM200). Then, polytetrafluoroethylene (PTFE, DuPont, Teflon 8A) binder was later added to the manually mixed, shaker-milled, or high-energy ball-milled mixture, such that the cathode film consists of the active compounds, carbon black, and PTFE in the weight ratio of 70:20:10. The components were then rolled into a thin film inside an argon-filled glove box. To assemble a cell for all cycling tests, 1 M $LiPF_6$ in 1:1 (volume ratio) ethylene carbonate (EC) and dimethyl carbonate (DMC) (BASF), Glass microfiber filters (Whatman), and Li metal foil (FMC) were used as the electrolyte, the separator, and the counter electrode, respectively. Coin cells (CR2032) were assembled in an argon-filled glove box and tested on a Maccor 2200 or an Arbin battery cycler at room temperature in the galvanostatic mode otherwise specified. The loading density of the cathode film was ~7 mg cm$^{-2}$. The specific capacity was calculated from the amount of the active compounds (70 wt%) in the cathode film.

**Characterization**. X-ray diffraction patterns for the as-prepared compounds were collected on a Rigaku MiniFlex diffractometer (Cu source) in the $2\theta$ range of 5–85°. Rietveld refinement was completed with using PANalytical X'pert HighScore Plus software. Elemental analysis on the compounds by Luvak Inc. was performed with direct current plasma emission spectroscopy (ASTM E 1097-12) for lithium, nickel, titanium and molybdenum, and with an ion selective electrode (ASTM D1179-10) for fluorine. SEM images were collected on Zeiss Gemini Ultra-55 Analytical Field Emission SEM in the Molecular Foundry at Lawrence Berkeley National Laboratory (LBNL). For the TEM sampling, particles were sonicated with ethanol and drop casted on an ultrathin carbon grid. Scanning TEM/EDS were acquired from a part of particles on JEM-2010F equipped with X-max EDS detector in the Molecular Foundry at LBNL. STEM/Electron energy loss spectra were acquired from JEM-ARM200F equipped with GIF Quantum spectrometer using dispersion of 0.25 eV in 200 keV.

**Solid-state nuclear magnetic resonance spectroscopy**. All $^{19}F$ NMR data were acquired at room temperature on a Bruker Ascend 400 MHz (9.4 T) DNP-NMR spectrometer, at a Larmor frequency of −376.6 MHz. The data were obtained under 30 kHz magic angle spinning using a 2.5 mm triple-resonance probe. All $^{19}F$ chemical shifts were referenced against polytetrafluoroethylene (PTFE, $\delta_{iso}$ = −121 ppm). $^{19}F$ spin echo spectra were acquired on LiF and as-synthesized LNF15 using a 90° RF pulse of 2.8 μs and a 180° RF pulse of 5.6 μs at 100 W. Recycle delays of 30 s and 50 ms were used for LiF and LNF15, respectively. In addition, a $^{19}F$ projected Magic-Angle Turning Phase-Adjusted Sideband Separation[22] (pj-MATPASS) isotropic spectrum was acquired on LNF15 using a 90° RF pulse of 2.8 μs at 100 W and a recycle delay of 50 ms. A $^{19}F$ probe background spin echo spectrum, acquired under the same conditions as the spin echo spectrum of LNF15 but on an empty 2.5 mm rotor, revealed no significant background signal. Lineshape analysis was carried out using the SOLA lineshape simulation package within the Bruker Topspin software.

**Differential electrochemical mass spectrometer measurement**. A DEMS was used to identify and quantify oxygen and carbon dioxide evolved during charging and discharging (Fig. 6 and Supplementary Figs. 4 and 5). The custom-built DEMS and the cell geometry used is described in previous publications[39–41]. The electrochemical cells used with the DEMS device were prepared in a dry argon glove box (<1 ppm $O_2$ and $H_2O$, MBraun USA, Inc.) using modified Swagelok design and the same materials as discussed previously. The assembled cells were charged under a static head of positive argon pressure (~1.2 bar) after being appropriately attached to the DEMS. Throughout the charge, argon gas pulses periodically swept accumulated gases to a mass spectrometer chamber. The mass spectrometer absolute sensitivity has been calibrated for $CO_2$ and $O_2$, and therefore the partial pressures of these gases can be determined. The amount of $CO_2$ and $O_2$ evolved is then quantified based on the volume of gas swept to the mass spectrometer per pulse.

**In situ X-ray diffraction**. For in situ XRD, an in situ cell was designed with a Be window for X-ray penetration. The cell was configured with a LN20 or LNF15 electrode film as the working electrode, Li metal foil as the counter electrode, 1 M $LiPF_6$ in EC:DMC (1:1, volume ratio) as the electrolyte, and glass fiber as the separator. Galvanostatic charge–discharge of the cells was performed on a Solartron electrochemical potentiostat (SI12287) between 1.5 and 4.6 V at 10 mA g$^{-1}$. The XRD patterns were obtained in 1 h intervals from a Bruker D8 Advanced Da Vinci diffractometer (Mo-source) in the $2\theta$ range of 7–36°. Rietveld refinement on the in situ XRD was performed using PANalytical X'pert HighScore Plus software for every other scan.

**Soft X-ray absorption spectroscopy**. Soft X-ray absorption spectroscopy was performed in the iRIXS endstation at Beamline 8.0.1 of the Advanced Light Source in LBNL[42]. Electrode samples were loaded into the ultra-high vacuum sXAS characterization chamber with a special sample transfer kit to avoid any air exposure effects. The undulator and spherical grating monochromator supplied a linearly polarized photon beam with a resolving power of up to 6000. The experimental energy resolution is about 0.15 eV. Experiments were performed at room temperature and with the linear polarization of the incident beam 45° from electrode surfaces. The sXAS spectra were collected simultaneously through both TEY mode with the probing depth around 10 nm and TFY mode with the probing depth larger than 100 nm, with the excitation X-ray beam hitting the same spot. All the spectra were normalized to the beam flux measured by the upstream gold mesh.

**Data availability**. The data sets generated and analyzed during the current study are available from the corresponding author on reasonable request.

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

## Acknowledgements

Work by J.L., R.J.C., and D.-H.K. was supported by Robert Bosch Corporation and Umicore Specialty Oxides and Chemicals. J.K.P. acknowledges NSF Graduate Research Fellowship (Grant No. DGE-1106400). S.S. acknowledges the support of the Advanced Light Source Doctoral Fellowships in Residence. B.D.M. gratefully acknowledges support from the Assistant Secretary for Energy Efficiency and Renewable Energy, Vehicle Technologies Office, of the U.S. Department of Energy under Contract No. DEAC02-05CH11231, under the Advanced Battery Materials Research (BMR) Program. Work at the Advanced Light Source is supported by DOE Office of Science User Facility under contract no. DE-AC02-05CH11231. Work at the Molecular Foundry was supported by the Office of Science, Office of Basic Energy Sciences, of the U.S. Department of Energy under contract No. DE-AC02-05CH11231. The authors would like to thank Dr. Jerry Hu for help with the NMR experiments, Daniil A. Kitchaev for valuable discussion, and would like to acknowledge the California NanoSystems Institute (CNSI) at the University of California Santa Barbara for experimental time on the 400 MHz NMR spectrometer.

## Author contributions

G.C. and J.L. planned the project. G.C. supervised all aspects of the research. J.L. designed, synthesized, characterized (XRD, in situ XRD), and electrochemically tested the proposed compounds. J.K.P. acquired and analyzed DEMS data with input from B.D.M. R.J.C. acquired and analyzed solid-state NMR data. S.S and W.Y. acquired and analyzed soft XAS data. D.-H.K. performed TEM-EDS and EELS. T.S. performed SEM. The manuscript was first written by J.L. and was revised by G.C., R.J.C., J.K.P., B.D.M., and W.Y. All authors contributed to discussions.

## Additional information

**Competing interests:** The authors declare no competing financial interests.

