## [Peer Review File · Nature Communications]

Reviewers' comments:

Reviewer #1 (Remarks to the Author):

This manuscript provides the fluorine-substituted Li-Ni-Ti-Mo oxides with a cation-disordered rocksalt structure for lithium-ion batteries. Reaction mechanisms are well studied for the fluorine-substituted sample. Direct synthesis of the fluorine-substituted sample is surely interesting. Nevertheless, further systematic studies are needed if the authors claim that oxygen loss is effectively reduced for the fluorine substitution. Surprising results are the contribution of Ni for the redox process is very small, unlikely $\text{LiNi}_{1/2}\text{Mn}_{1/2}\text{O}_2$. Specific points are described as follows;

1. A new peak at 528.5 eV is assigned as the formation of hole in non-bonding oxygen 2p band, but the trend is different from the Li_2MnO_3 phase as recently reported by Bruce et al. Energy level of non-bonding oxygen 2p is expected to be the same for Ni, Ti and Mn. Or does Ti/Mo substitution change the energy level of non-bonding oxygen 2p? Can you provide the theoretical data for O K-edge spectra as claimed in the manuscript?
2. Mo L-edge was not shown in this manuscript. However, Mo L-edge is a good indicator for the oxygen loss (see DOI: 10.5796/electrochemistry.84.797 written by Yabuuchi et al., Li-Fe-Mo-O system). Oxygen loss results in the reduction of Mo^{6+} on discharge.
3. If authors claim that the oxygen loss is suppressed for the fluorine-substitution, O K-edge spectra for LN20 or LN15 are needed for comparison.
4. Rietveld analysis, oxygen and fluorine occupancies seem to be refined. However, no meaning by X-ray diffraction. We cannot distinguish O and F by X-ray diffraction. Or did you use neutron diffraction?

Minor things,

1. The preparation of shaker milled LNM15 is not described in the manuscript.
2. Did you check F_2 gas generation on charge?
3. Please show us the data of F K-edge if you have collected.
4. Figure S6, caption, (c) and (d) must be (a) and (b).

Reviewer #2 (Remarks to the Author):

This work reports the inhibition of lattice oxygen loss for disordered rock-salt materials by partial fluorine substitution. It is interesting that the $\text{Ni}^{4+/2+}$ redox reaction is useful in the cation-disordered materials with high cell voltage and low polarization, when the anion sublattice has mixed O and F.

1. Page 4, line 1 to 3: what is the different between the solid-state synthesis method in this work and the conventional solid-state synthesis?
2. Page 5, line 5 from bottom: The partial substitution of F for O in the rock-salt phase resulted in decrease in polarization. This is reasonable, as has been reported previously for vanadium-based materials. Oxygen loss has been observed in this work. How about fluorine loss for the mixed O/F materials?
3. What are other possible reasons for the capacity loss?
4. Redox reaction of O anion has been reported previously for Nb-based materials, which contributed to additional capacity. What is the final Li content at charged state? How does the Li content change when the O redox proceeds? Do you expect any structural change for the deep delithiated phase?

5. To the best of my knowledge, this paper showed for the first time that low polarization of Ni⁴⁺/Ni²⁺ redox reactions has been observed in disordered rock-salt materials. The typical two-plateau discharge curves (as also observed in Li₂NiTiO₄) evolved into one slope, after F substitution. Some citations should be added for related work.

6. Through fluorine substitution (Figure 4), the change in the discharge voltage profiles was mainly in the low-voltage range (below 3.5 V), related to Ni³⁺/Ni²⁺ redox reaction. The high voltage range (with a capacity of about 70 mAh g⁻¹ above 3.5 V) did not change, suggesting Ni⁴⁺/Ni³⁺ should occur. However, from the XAS analysis, only Ni³⁺/Ni²⁺ were observed. How to understand this?

Minor points:

1. The discharge energy density mentioned in the introduction part (Page 3, line 9) is 800 Wh/kg (or 3380 Wh/L), which is different from that (790 Wh/kg, or 3330 Wh/L) in the discussion part (Page 12, second paragraph).

2. Page 13, line 4: the composition for LNF 15 is wrong.

3. Page 16: Ref. 8 and Ref. 12 are the same.

4. Page S2, Caption for Table S1, line 3: the composition for LNF15 should be O1.85F0.15, not O0.185F0.15.

REVIEWERS' COMMENTS:

Reviewer #1 (Remarks to the Author):

Well updated.

Reviewer #2 (Remarks to the Author):

The authors have addressed properly the issues in the revised version. The manuscript has been improved largely. I recommend for publication at the current form.

Response to Reviewer #1

This manuscript provides the fluorine-substituted Li-Ni-Ti-Mo oxides with a cation-disordered rocksalt structure for lithium-ion batteries. Reaction mechanisms are well studied for the fluorine-substituted sample. Direct synthesis of the fluorine-substituted sample is surely interesting. Nevertheless, further systematic studies are needed if the authors claim that oxygen loss is effectively reduced for the fluorine substitution. Surprising results are the contribution of Ni for the redox process is very small, unlikely $\text{LiNi}_{1/2}\text{Mn}_{1/2}\text{O}_2$. Specific points are described as follows;

1. A new peak at 528.5 eV is assigned as the formation of hole in non-bonding oxygen 2p band, but the trend is different from the Li_2MnO_3 phase as recently reported by Bruce et al. Energy level of non-bonding oxygen 2p is expected to be the same for Ni, Ti and Mn. Or does Ti/Mo substitution change the energy level of non-bonding oxygen 2p? Can you provide the theoretical data for O K-edge spectra as claimed in the manuscript?

First of all, we greatly appreciate the reviewer 1's constructive comments on our work. When it comes to comment #1, we would like to point out that the 528.5 eV peak was not assigned to O-hole formation in non-bonding O 2p band. Instead, we assigned it to Ni oxidation (Page 9, line 257–258 in the original manuscript). O-hole creation was assigned to the intensity gain between 530–535 eV (especially at ~532 eV), which is consistent with the observation by Bruce *et al.* (Page 10, line 261–264, in the original manuscript). We revised our manuscript to explain our interpretation more clearly, and hope that this clarifies the issue.

2. Mo L-edge was not shown in this manuscript. However, Mo L-edge is a good indicator for the oxygen loss (see DOI: 10.5796/electrochemistry.84.797 written by Yabuuchi et al., Li-Fe-Mo-O system). Oxygen loss results in the reduction of Mo^{6+} on discharge.

Detecting Mo reduction could indeed be an indirect way of proving oxygen loss. However, our DEMS experiment is the most advanced and direct technique to show when and how much oxygen loss occurs from our materials. In addition, the energy range for the Mo L-edge is beyond the detectable range of our soft XAS measurements.

3. If authors claim that the oxygen loss is suppressed for the fluorine-substitution, O K-edge spectra for LN20 or LN15 are needed for comparison.

O K-edge spectra are only a very indirect way of proving oxygen loss, as the K-edge spectra relate to oxygen redox and changes in oxygen hybridization that is left in the sample, but cannot detect oxygen that has been lost. In contrast, DEMS directly measures the oxygen lost, and indicates that less oxygen is lost from LNF15 than from LN15 or LN20.

In this work, O loss from LNF15 is reduced as a result of the “increased Ni content” upon fluorination, and not because fluorine affects the O 2p states. In fact, fluorine being outside of the first coordination shell of oxygen, does not directly affect the O 2p states. As clearly

evidenced by the DEMS results in Fig. 6 of the paper and Reply Fig. 1 of this response letter, the amount of O loss ($\text{LNF15} < \text{LN15} < \text{LN20}$) decreases with increasing Ni^{2+} content (Ni redox reservoir). By lowering the average anion valence, fluorination allows for a higher Ni content in the Li–Ni–Ti–Mo–O system without sacrificing the Li-excess level. We have revised our draft to clarify this point.

As discussed by Yabuuchi *et al.*, *Nature Commun.* (2016), changing metal species is another way to control O loss, since it directly affects the M d –O $2p$ hybridization and O $2p$ states, making the O K-edge comparison relevant. However, our work does not target a modification of the O $2p$ states by changing metal species, as our different samples contain the same metals. Therefore, we believe that a comparison of the O K-edge spectra is not directly relevant and beyond the scope of this paper, especially when we have DEMS data that clearly supports our claims that higher Ni content (either for lower Li-excess level: $\text{LN15} > \text{LN20}$, or for fluorination: $\text{LNF15} > \text{LN15}$) leads to less oxygen loss.

4. Rietveld analysis, oxygen and fluorine occupancies seem to be refined. However, no meaning by X-ray diffraction. We cannot distinguish O and F by X-ray diffraction. Or did you use neutron diffraction?

We first refined the metal occupancy. Then, we fixed that to refine O/F occupancies to check if profile fitting can be improved, which however led to negligible changes of the occupancies from their initial values set based on elemental analysis results. Nevertheless, as the reviewer 1 points out, distinguishing O and F by XRD is very difficult, which is the reason why we performed solid-state NMR and EDS to prove F-substitution in the bulk lattice of LNF15. We have clarified this point in the revised supplementary information.

Minor 1. The preparation of shaker milled LNM15 is not described in the manuscript.

We have improved our description on the preparation of shaker-milled LNF15 in the revised manuscript.

Minor 2. Did you check F_2 gas generation on charge?

In Reply Fig. 1 of this response letter, we present the ion current (per mass of active compounds, A/mg) obtained from O_2 and F_2 evolution from LN15, LN20, and LNF15 upon first charge and discharge. F_2 signal for LNF15 is silent as is the case for LN15 and LN20, indicating that no F_2 gas is evolved from LNF15. We have included this data as the supplementary Fig. 5.

Minor 3. Please show us the data of F K-edge if you have collected.

We did not collect the F K-edge spectra from soft XAS because our electrode film is made of F-containing PTFE binder, which shows a strong F signal overlapping with the F signal from the cathode.

Minor 4. Figure S6, caption, (c) and (d) must be (a) and (b).

We have corrected the Fig. S6 caption.

Reply figure 1 Ion current per mg of active compounds (A/mg), from O₂ and F₂ gas evolution upon initial charge and discharge of (a) LN15, (b) LN20, and (c) LNF15, between 1.5 and 4.8 V at 20 mA/g.

Response to Reviewer #2

This work reports the inhibition of lattice oxygen loss for disordered rock-salt materials by partial fluorine substitution. It is interesting that the Ni^{4+/2+} redox reaction is useful in the cation-disordered materials with high cell voltage and low polarization, when the anion sublattice has mixed O and F.

1. Page 4, line 1 to 3: what is the different between the solid-state synthesis method in this work and the conventional solid-state synthesis?

First of all, we greatly appreciate the reviewer 2's positive comments and thoughtful questions. For the comment 1, there is no difference in the solid state method itself. We wanted to point out that (i) solid state synthesis has failed to incorporate fluorine into "layered" rocksalt materials, and that (ii) fluorination to "disordered" rocksalt materials had been achieved only by mechanochemical ball milling methods. Our work shows for the first time that some level of fluorination can be achieved in disordered rocksalt materials using a standard solid-state method. We have revised our manuscript to clarify these points.

2. Page 5, line 5 from bottom: The partial substitution of F for O in the rock-salt phase resulted in decrease in polarization. This is reasonable, as has been reported previously for vanadium-based materials. Oxygen loss has been observed in this work. How about fluorine loss for the mixed O/F materials?

As shown in Reply Fig. 1, there is no F_2 gas evolution from LNF15. The ion current signal from F_2 gas-evolution is silent for LNF15, as for LN15 and LN20. We have included this data as the supplementary Fig. 5.

3. What are other possible reasons for the capacity loss?

Reply figure 2 The initial two-cycle voltage profiles of LNF15 when it is cycled between (a) 1.5–4.5 V and (b) 1.5–5.0 V at 30 mA/g. (c) The (002) XRD-peak of LNF15 before cycling, after two cycles between 1.5–4.5 V, and between 1.5–5.0 V. (d) As prepared separator, separator after two cycles between 1.5–4.5 V, separator after two cycles between 1.5–5.0 V.

Capacity fading of LN15, LN20 and LNF15 can come from different reasons. Firstly, electrolyte decomposition at a high voltage (> 4.5 V) can create resistive surface layers on the cathode particles, as has been demonstrated for a number of materials [Xu *et al.*, *Chem. Rev.*, 2014; Hong *et al.*, *Chem. Mater.*, 2012]. In addition, electrolyte decomposition can lower the ionic conductivity of the electrolyte, hence the cell impedance grows. The combination of these two

effects explains why capacity retention of LNF15 is improved when the charge cutoff voltage is decreased from 4.6 V (Fig. 4c) to 4.5 V (Supplementary Fig. 9c). Secondly, capacity fading may come from metal dissolution from the materials upon cycling, as can be inferred from the discoloration of a separator in a cycled cell (Reply Fig. 2d). This dissolution issue for LN15, LN20, and LNF15 is under careful investigation and will be presented in a separate paper. One thing that is clear at this point is that there is Ni dissolution (consistent with the greenish coloration (Ni^{2+}) of separator), which is a known issue of Ni^{2+} -containing cathode compounds.

4. Redox reaction of O anion has been reported previously for Nb-based materials, which contributed to additional capacity. What is the final Li content at charged state? How does the Li content change when the O redox proceeds? Do you expect any structural change for the deep delithiated phase?

For the 1.5–4.5 V 20 mA/g cycling test (Supplementary Fig. 8c), where electrolyte decomposition (hence capacity from side reactions) is expected to be minimal based on DEMS results (Fig. 6), the final Li content in LNF15 can be computed from the overall charge capacity (239 mAh/g, first charge), giving 0.34 Li/f.u. left at the top of the first charge. However, the final Li content is expected to change with the charge cut-off voltage and the rate of cycling. Due to partial overlap between the Ni and O redox in the Li–Ni–Ti–Mo–O(F) system [Seo *et al.*, *Nature Chem.*, 2016], the Li-content range for O redox is not clearly defined in LNF15.

While oxygen loss is much reduced in LNF15, as compared with LN15 or LN20, the material also loses oxygen upon charge above 4.5 V. O loss from LNF15 becomes more severe with increasing cut-off voltage, which leads to formation of a cation-densified surface structure with poorer 0-TM percolation (lowered Li excess content), hence slower Li diffusion, in highly delithiated LNF15. This explains why polarization is larger in the 1.5–5.0 V cycling curves of LNF15 than in the 1.5–4.5 V curves, as can be seen from Reply Fig. 2a and 2b.

In addition, we find that a shift of the (002) XRD peak of LNF15 to a lower angle is larger after two cycles between 1.5–5.0 V than between 1.5–4.5 V (Reply Fig. 2c). This result indicates an expansion of the disordered LNF15 lattice after cycling, which becomes more pronounced after LNF15 goes through deeper delithiation: $a = 4.1428 \text{ \AA}$ (before cycling), 4.1522 \AA (1.5–4.5 V), and 4.1559 \AA (1.5–5.0 V). This lattice expansion can be explained by an irreversible cation rearrangement upon delithiation, which leads to a metastable LNF15 disordered structure upon lithium reinsertion, with a larger volume than that of the as-prepared LNF15 structure.

5. To the best of my knowledge, this paper showed for the first time that low polarization of $\text{Ni}^{4+/2+}$ redox reactions has been observed in disordered rock-salt materials. The typical two-plateau discharge curves (as also observed in $\text{Li}_2\text{NiTiO}_4$) evolved into one slope, after F substitution. Some citations should be added for related work.

As the reviewer 2's comment, this paper indeed shows for the first time low polarization in Ni-containing disordered compounds, which was the primary goal of this research. To emphasize this point, we discussed related work and added citations in the revised draft.

6. Through fluorine substitution (Figure 4), the change in the discharge voltage profiles was mainly in the low-voltage range (below 3.5 V), related to Ni³⁺/Ni²⁺ redox reaction. The high voltage range (with a capacity of about 70 mAh g⁻¹ above 3.5 V) did not change, suggesting Ni⁴⁺/Ni³⁺ should occur. However, from the XAS analysis, only Ni³⁺/Ni²⁺ were observed. How to understand this?

First of all, we did not claim that there is no Ni³⁺/Ni⁴⁺ redox at all. We argued that Ni²⁺/Ni⁴⁺ reservoir (Ni²⁺/Ni³⁺/Ni⁴⁺) is only partially utilized in LNF15, which was also observed for LN20 (Lee *et al.*, *Energy Environ. Sci.*, 2015; Seo *et al.*, *Nature Chem.*, 2016). Secondly, we would like to be cautious and not directly relate discharge profiles to redox mechanism, because many features, either chemical, variations in short range order, or different degrees of polarization can all change the discharge profile even when the overall redox mechanism is similar. For all of the materials considered in this study, the first charge consists of (i) partial Ni²⁺/Ni^{3+,4+} oxidation, (ii) O oxidation, and (iii) O loss, these three processes occurring to a different extent in LN15, LN20 and LNF15. Depending on the amount of O loss, the discharge profile can change a lot, even when the main redox mechanisms are O and Ni reduction in the bulk.

It is important to emphasize that the low voltage discharge capacity, especially at ~2.2 V (corresponding to the well-defined plateau in the LN15 or LN20 electrochemical curves), should not be directly related to the Ni²⁺/Ni³⁺ redox capacity. This is because the 2.2 V-voltage is attributed to the Mo⁶⁺ or Ti⁴⁺ reduction near the particle surface after O loss has occurred in the first charge (Lee *et al.*, *Energy Environ. Sci.*, 2015). The low voltage plateau is more clearly observed for LN15 or LN20 than for LNF15, because the larger amount of O loss in the pure oxides enhances Mo and Ti reduction at the surface upon discharge. In addition, substantial O loss from LN15 and LN20 leads to the formation of a cation-densified surface structure which reduces Li mobility, making the low voltage reactions (Mo/Ti reduction at the surface) more apparent on discharge (=> negative over-potential from mass-transfer resistance). Combining these two factors, we see an exaggerated low voltage (~2.2 V) plateau in LN15 or LN20, instead of observing higher potentials from O and Ni^{3+,4+}/Ni²⁺ reduction in the bulk. Since O loss is mitigated in LNF15, the low voltage plateau is much less pronounced and instead we observe a higher voltage that is closer to the potential at which O and Ni are reduced. We believe that this is as far as we can go in terms of the analysis of the voltage profiles.

Minor 1. The discharge energy density mentioned in the introduction part (Page 3, line 9) is 800 Wh/kg (or 3380 Wh/L), which is different from that (790 Wh/kg, or 3330 Wh/L) in the discussion part (Page 12, second paragraph).

The 800 Wh/kg was derived from the 1.5–4.6 V 10 mA/g cycling test (Fig. 5c), and the 790 Wh/kg was derived from the 1.5–4.6 V 20 mA/g cycling test (Fig. 4d). We revised our draft to prevent this confusion.

Minor 2. Page 13, line 4: the composition for LNF 15 is wrong.

We have corrected the composition.

Minor 3. Page 16: Ref. 8 and Ref. 12 are the same.

We have changed reference 12.

Minor 4. Page S2, Caption for Table S1, line 3: the composition for LNF15 should be O1.85F0.15, not O0.185F0.15.

We have corrected the composition.

Response to Reviewer #1

Well updated.

We greatly appreciate the reviewer 1's positive comment on our revised manuscript.

Response to Reviewer #2

The authors have addressed properly the issues in the revised version. The manuscript has been improved largely. I recommend for publication at the current form.

We greatly appreciate the reviewer 2's positive comment on our revised manuscript.